# State Anxiety after Exergame Beach Volleyball Did Not Differ between the Single and Multiplayer Modes in Adult Men

**DOI:** 10.3390/ijerph182010957

**Published:** 2021-10-18

**Authors:** Vinnycius de Oliveira, Ricardo Viana, Naiane Morais, Gustavo Costa, Marilia Andrade, Rodrigo Vancini, Claudio de Lira

**Affiliations:** 1Faculdade de Educação Física e Dança, Universidade Federal de Goiás, Goiânia 74690-900, GO, Brazil; vinnyciusnunes@discente.ufg.br (V.d.O.); morais.nsm@gmail.com (N.M.); conti02@ufg.br (G.C.); 2Escola Superior de Educação Física e Fisioterapia de Goiás, Universidade Estadual de Goiás, Goiânia 74075-110, GO, Brazil; vianaricardoborges@hotmail.com; 3Faculdade Estácio de Sá de Goiás, Goiânia 74063-010, GO, Brazil; 4Departamento de Fisiologia, Universidade Federal de São Paulo, São Paulo 04021-001, SP, Brazil; marilia1707@gmail.com; 5Centro de Educação Física e Desportos, Universidade Federal do Espírito Santo, Vitória 29075-810, ES, Brazil; rodrigoluizvancini@gmail.com

**Keywords:** mental health, physical activity, physical exercise, active video game

## Abstract

This study compared the exergame beach volleyball’s acute effects on state anxiety level in single vs. multiplayer mode in adult men. Sixty adult men (age: 21.98 [4.58] years, body mass: 75.40 [15.70] kg, height: 1.77 [0.09] m, and body mass index: 24.19 [5.44] kg/m^2^; data are expressed as median [interquartile range]) were assigned to play exergame of beach volleyball in single- or multiplayer mode for approximately 30 min using the Xbox 360 Kinect^®^. The state anxiety level was evaluated before and after the intervention. There was no significant difference in the state anxiety levels after an exergame session between the single and multiplayer modes (*p*-value = 0.407, effect size (r_B_) = −0.12, defined as small). Furthermore, there was no significant difference in the state anxiety levels before and after an exergame session in single-player mode (*p*-value = 0.516, effect size (d) = 0.14, defined as trivial) and multiplayer mode (*p*-value = 0.053, r_B_ = 0.43, defined as medium). In conclusion, state anxiety level after exergame beach volleyball did not differ between the single and multiplayer modes in adult men.

## 1. Introduction

Anxiety is a normal reaction of the body to stress, but it can become a mental health disorder at increased levels. In the latter case, symptoms of anxiety impair the quality of life [1]. Anxiety is defined as a state of excessive concern, anticipation, suffering, and panic [2]. Anxiety is classified into state anxiety (transitory feelings; reflecting a current emotional state) and trait anxiety (chronic; reflecting a more stable feature) [3]. 

According to the World Health Organization, 264 million people worldwide had anxiety disorders in 2015, corresponding to 3.6% of the world population [4]. In 2015, Brazil had the highest prevalence of anxiety disorders globally with approximately 19 million people (9.3% of the Brazilian population) [4]. Physical exercise is an effective tool to prevent and treat anxiety [5]. Moreover, physical inactivity is associated with mental health disorders [6].

Despite the benefits of regular physical exercise, most people are physically inactive [7]. People attribute this high level of physical inactivity to their lack of interest in exercising among other barriers, such as lack of time, company, adequate climate, motivation, and enjoyment; fear of injury and leaving home unattended; involvement of high financial costs (for example, monthly fitness center fee); and presence of physical tiredness [8,9,10,11,12]. To improve this situation, new modalities of physical exercises have emerged such as the exergames. Exergames are a new generation of videogames that come under the category of active video games [13]. The exergames can help overcome some of the barriers associated with reduction in performing traditional physical activity, such as lack of adequate climate and fear of leaving home unattended, because the activity can be performed at home. Another barrier for reduced participation in physical activity is the widespread view that exercise is not enjoyable. In this regard, it has been suggested that exergames may be more enjoyable than traditional training programs [14].

The exergames, similar to traditional video games, can also be played in single or multiplayer modes. In the multiplayer mode, individuals strive to play more, as playing in a team (doubles) causes more excitement, promotes social interaction, and involves greater energy expenditure [14,15,16]. Peng and Crouse (2013) investigated the effects of a minigame Space Pop performed in the exergame Kinect Adventures^®^ on the enjoyment levels and on a possibility of future engagement between single and multiplayer modes and found that the multiplayer mode presented a higher probability of future engagement and enjoyment than the single-player mode [16]. Barkman et al. (2016) investigated the effects of several exergames performed by using the Xbox 360 Kinect^®^ in young people on energy expenditure between single and multiplayer modes [17]. They found that the multiplayer mode caused higher energy expenditure than the single-player mode.

However, to the best of our knowledge, there is no study comparing the acute effects of single vs. multiplayer modes on state anxiety levels. Viana et al. (2017) showed that a single session of the exergame Zumba^®^ Fitness in the Xbox 360 Kinect^®^ of moderate intensity in the single-player mode decreased the state anxiety level [18]. Recently, Morais et al. (2021) compared state anxiety levels between a dance exergame session and a traditional aerobic exercise [19]. The authors showed that dance exergames might be used as a tool to reduce anxiety level. Thus, it is reasonable to assume that the multiplayer mode would present similar or higher reduction. Therefore, the present study aimed to compare the exergame Kinect Sports^®^ beach volleyball’s acute effects on state anxiety levels, enjoyment levels, and exercise intensity in single vs. multiplayer modes in healthy adult men. Previous studies showed a decrease in state anxiety levels after one session of traditional exercise and studies with exergames showed better outcomes for enjoyment levels, possibility of future engagement, and energy expenditure in the multiplayer mode [16,17,20]. Hence, we hypothesized that the multiplayer mode would provide a more considerable decrease in state anxiety levels than the single-player mode, as participants find the multiplayer mode more exciting. We also hypothesized that the multiplayer mode would provide higher enjoyment and exercise intensity than the single-player mode.

## 2. Materials and Methods

### 2.1. Participants 

Sixty-two healthy adult men volunteered to participate in this study (a convenience sample) (Table 1). The participants were undergraduate students from the Faculty of Physical Education and Dance from the Federal University of Goiás (Brazil). The inclusion criteria were male sex and age between 18 and 40 years. The exclusion criteria were as follows: (i) having answered “yes” to one or more questions of the Physical Activity Readiness Questionnaire (Physical Activity Readiness Questionnaire—PAR-Q, 2002) and (ii) use of any type of psychotropic drug. The participants were not clinically diagnosed with anxiety, depression, and/or other mental health disorder as per self-report by the participants about their current status. No extra course credit was offered for their participation. Forty-four participants (~73%) were physically active (self-reported). Thirty-two participants were allocated to the single-player mode, and 30 participants were assigned to the multiplayer mode. Two participants from the single-player mode discontinued and were excluded. Participants were asked to bring a friend to the laboratory session for the multiplayer mode (who had a pre-existing relationship with the participant before participation). Informed consent was obtained from all participants included in the study. All experimental procedures were approved by the Federal University of Goiás Ethics Committee (approval number: 89043718.4.0000.5083) and followed the principles outlined in the Declaration of Helsinki. The flow diagram of the study is presented in Figure 1.

### 2.2. Study Design 

This was a between-group study design composed of two visits. At the first visit, the participants were submitted to anamnesis and anthropometric, cardiorespiratory fitness, and anxiety trait assessments. All participants were familiarized with the exergame Kinect Sports^®^ beach volleyball. At the second visit, the participants performed a single session of the exergame Kinect Sports^®^ beach volleyball in single or multiplayer mode against another player for approximately 30 min. At the end of each match, heart rate (HR) and rating of perceived exertion (RPE) were measured to characterize the exercise intensity. This measurement aimed to investigate the possible influence of exercise intensity on the state anxiety levels. State anxiety levels were assessed before and after the exergame session. The enjoyment level was evaluated after the exergame session. 

### 2.3. Experimental Procedures

#### 2.3.1. Anamnesis and Anthropometric Assessment

The anamnesis consisted of administration of the Physical Activity Readiness Questionnaire. This questionnaire comprised seven questions evaluating the participants’ general health condition and fitness to perform an exercise. If a participant answered “yes” to one or more questions, the participant was excluded from the study. Body mass was measured using a digital balance (Omron, HN-289, Lake Forest, IL, USA), and height was measured using a wall stadiometer (Caumaq, Brazil). Body mass index was calculated by dividing body mass by height squared (kg/m^2^). 

#### 2.3.2. Cardiorespiratory Fitness Assessment

Participants were made to perform the Ebbeling test (1991), performed on a motorized treadmill (ATL, Inbramed, Brazil), to predict maximal oxygen uptake ($\dot{V}$O_2_max) [21]. The Ebbeling test consists of two 4 min stages. At the first 4 min of the protocol, participants walked at 0% slope and at a speed corresponding to HR range from 50% to 70% of the maximal HR predicted for their age. Then, for the last 4 minutes of the test, the treadmill slope was increased to 5% while the walking speed remained the same. HR was measured at the end of the test. The following equation estimated participants’ $\dot{V}$O_2_max:
(1)$$\dot{V}O_{2}\max=15.1+21.8(\mathrm{speed})-0.327(\mathrm{HR})-0.263(\mathrm{speed}\times \mathrm{age})+0.00504(\mathrm{HR}\times \mathrm{age})+5.98\times \mathrm{sex}.$$
where speed was expressed in miles per hour, HR in beats per minute (bpm), age in years, and sex as 0 for females or 1 for males.

#### 2.3.3. Exergame Session

The console used was the Xbox 360 (Microsoft^®^, Washington, DC, USA). This console allows a connection to a movement sensor, the Kinect^®^ (Microsoft, Washington, DC, USA). This sensor enables the player to interact with the videogame without the necessity of a remote control and/or joysticks. The exergame Kinect Sports^®^ included six sports modalities (beach volleyball, soccer, track and field, bowling, boxing, and table tennis). In the current study, the participants played beach volleyball during approximately 30 min. In this period, participants played around six matches of beach volleyball. Between each match, there was a period of 1–2 min in order to participant select the option to play another match on the videogame menu.

We chose beach volleyball because it is a common sport among Brazilians. This exergame is inherently competitive in both single and multiplayer modes, with a possible win or lose outcome in both conditions. Each visit was at an interval of 24–72 h. All participants visiting the laboratory wore the appropriate outfit for physical exercise; refrained from eating two hours before exercising; and abstained from caffeine, alcohol, and strenuous physical activity on the day of the experiment. The temperature in the laboratory ranged from 21 °C to 23 °C. An experienced researcher supervised each participant. 

#### 2.3.4. State Anxiety Level Assessment

The state anxiety level was evaluated using the 20-item state anxiety component from the State-Trait Anxiety Inventory [22]. The participants answered each of the questions according to how they felt “right now, that is, at this moment.” The state anxiety scoring system is based on a 4-point Likert scale (1 = not at all, 2 = somewhat, 3 = moderately so, and 4 = very much so). Overall scores can vary from 20 (minimum) to 80 (maximum). A score ≤30 indicates a low level of state anxiety; from 31 to 49, an intermediate level of state anxiety; and ≥50, a high level of state anxiety [22]. Participants answered the State-Trait Anxiety Inventory inside a sound-attenuated room. The State-Trait Anxiety Inventory was chosen because of its easy application and low cost.

#### 2.3.5. Exercise Intensity Assessment

Exercise intensity was monitored by measuring HR and RPE. HR was monitored using an HR monitor (H10, Polar, Finland). RPE was monitored using the Borg Scale (6–20) [23]. 

#### 2.3.6. Enjoyment Assessment

Enjoyment was assessed using the Physical Activity Enjoyment Scale by Kendzierski and DeCarlo (1991) modified by Graves et al. (2010) In brief, the modified Physical Activity Enjoyment Scale uses only five items from the original scale [24,25]. Participants rated the extent to which they agreed to each item on a 7-point Likert-type scale. Scores could range from a minimum of 5 (no enjoyment level) to a maximum of 35 (the highest level of enjoyment). 

### 2.4. Statistical Analysis

Shapiro–Wilk test was utilized to test the normality of data. Mann–Whitney U test was used to compare differences at baseline, enjoyment level after exergame session, and the change in state anxiety levels in single vs. multiplayer mode, as these data presented a non-normal distribution. Independent Student’s *t*-test was used to compare the mean HR, percentage of maximum HR, and RPE in single vs. multiplayer mode, as these data presented a normal distribution. Wilcoxon test was used to compare the state anxiety levels before and after the single and multiplayer modes. Effect size (Cohen’s d) was used for Student’s *t*-test [26]. According to Cohen, the d values were classified as “trivial” (d < 0.30), “small” (0.30 ≤ d < 0.50), “medium” (0.50 ≤ d < 0.80), and “large” (d ≥ 0.8) [27,28]. The effect size used for the Mann–Whitney U test and Wilcoxon test was the rank-biserial correlation (r_B_) [27,28]. r_B_ values classification was based on the Pearson’s correlation coefficient (r). The values were classified as “trivial” (r_B_ < 0.10), “small” (0.10 ≤ r_B_ < 0.30), “medium” (0.30 ≤ r_B_ < 0.50), and “large” (d ≥ 0.5) [27,28]. Parametric data were presented as mean ± standard deviation, mean difference, and 95% confidence interval (95% CI). Non-parametric data were presented as the median and interquartile range (IQR), median difference, and 95% CI. All data were analyzed through JASP (version 0.12.2, Amsterdam, The Netherlands). The level of significance was set at <0.05.

## 3. Results

### 3.1. State Anxiety

Both the single and multiplayer modes presented intermediate state anxiety levels before (mean: 37.33 ± 8.67 and median = 34.00 [IQR: 5.75], respectively) and after (mean: 36.10 ± 7.02 and mean: 32.76 ± 5.73, respectively) the exergame session. There was no significant difference in the change in state anxiety levels after the exergame session between the single and multiplayer modes (median difference = −2 [95% CI: −5; 2], *p*-value = 0.407; r_B_ = −0.12 “small” [95% CI: −0.39; 0.16]) (Figure 2A). Furthermore, there was no significant difference in the state anxiety levels before and after exergame session in the single-player mode (mean difference = 1 [95% CI: −2.5; 4.5], *p*-value = 0.516, r_B_ = 0.14 “small” [95% CI: −0.26; 0.50]) and multiplayer mode (median difference = 3[95% CI: 0; 7], *p*-value = 0.053; r_B_ = 0.43 “medium” [95% CI: 0.02; 0.72]) (Figure 2B). In addition, there was no difference in the baseline state anxiety levels between the single and multiplayer modes (*p*-value > 0.05).

### 3.2. Enjoyment Level

The single-player (median: 32.0 [IQR: 6.0]) and multiplayer (median: 32.5 [IQR: 5.0]) modes reported high enjoyment levels after the exergame session. However, there was no significant difference in enjoyment levels after the exergame session between the two modes (median difference = 0 [95% CI: −2; 2], *p*-value = 1.000, r_B_ = 0.00 “trivial” [95% CI: −0.28; 0.28]) (Figure 3). 

### 3.3. Exercise Intensity 

There was no significant difference in RPE between single (12 ± 2) and multiplayer (11 ± 2) modes (mean difference = −0.5 [95% CI: −1.4; 0.3], *p*-value = 0.201, d = −0.33 “small” [95% CI: −0.84; 0.18]) (Figure 4A). There was no significant difference in mean HR between single (119 ± 18 bpm) and multiplayer (116 ± 17 bpm) mode (mean difference = −3 bpm [95% CI: −12; 6], *p*-value = 0.501, d = −0.18 “trivial” [95% CI: −0.68; 0.33]) (Figure 4B). 

## 4. Discussion

The present study evaluated the effects of the exergame Kinect Sports^®^ beach volleyball on state anxiety levels, enjoyment level, and exercise intensity between single and multiplayer modes in healthy adult men. Contrary to our initial hypothesis, the state anxiety levels after the exergame session did not differ between single and multiplayer modes, and the exergame sessions did not improve the state anxiety level. Moreover, there were no differences in the enjoyment levels and exercise intensity between the single and multiplayer modes. 

We found no statistical difference in the changes of state anxiety levels between the single and multiplayer modes and no statistical difference before and after a session of exergame in both the modes. Recently, da Silva et al. (2021) conducted a study to compare the acute effects of an exergame-based calisthenics session versus a traditional calisthenics session on state anxiety levels in healthy adult men [29]. The authors did not find significant differences between interventions on state anxiety levels. One possible explanation for the current study results may be related to the light intensity of the exergame session evoked by the single and multiplayer modes. Previous studies demonstrated that moderate-intensity physical exercise provides a reduction in anxiety levels [29,30,31,32,33]. Viana et al. (2017) and Morais et al. (2021) evaluated the effects of a single session of the exergame Zumba Fitness^®^ performed at moderate intensity on the state anxiety level in healthy young women and found a significant reduction after the session. In addition to moderate-intensity exercise, the authors attributed the anxiolytic effect to the participants’ high enjoyment levels. In the present study, the enjoyment level was found to be high (approximately 32, corresponding to 89% of the relevant scale’s maximal score) [18,19]. Thus, high enjoyment levels seem to be insufficient in improving the state anxiety levels using exergames. Therefore, the significant decrease in state anxiety levels seen in the previous study might be dependent on the exergame session intensity. Furthermore, Viana et al. (2017) and Morais et al. (2021) recruited only women [18,19]. A previous study also demonstrated that females have a more positive psychosocial response to exergame sessions than males [14]. Our findings are relevant as our study findings, which show no significant difference in state anxiety levels between the single and multiplayer modes and no statistical difference before and after a session of exergame in a single or multiplayer mode, are seen in healthy adult men.

Ensari et al. (2015) reported that a bias, which permeates many studies on anxiety, is associated with recruiting persons with average or lower levels of state anxiety. Therefore, it is not possible to further decrease anxiety levels [20]. This situation is known as the floor effect. Individuals presenting with high baseline state anxiety levels showed a more significant decrease in state anxiety levels post-exercise [34]. In the present study, participants were healthy and demonstrated intermediate anxiety levels, considered as a normal state of anxiety. It may have contributed to the minor impact of the exergame sessions on the state anxiety levels. Moreover, as regular physical exercise provides a protective effect on mental health disorders, the participants’ high cardiorespiratory fitness in the present study may have influenced the decreased state anxiety levels after the exergame Kinect Sports^®^ beach volleyball [6,35,36,37,38,39]. Several studies elucidated the acute and chronic effects of traditional exercises, such as resistance and aerobic training, on state anxiety levels [20]. However, other studies neither found decreased state anxiety levels after a training session nor an anxiogenic effect [40,41]. Garwin et al. (1997) assessed the acute effect of aerobic exercise on the state anxiety levels. A total of 60 participants were recruited and performed resistance training, cycling, relaxation or control intervention of 50-min duration each. The results showed that the state anxiety levels reduced only in the control group and after relaxation, which were applied as interventions. However, the authors found a significant reduction in the state anxiety levels one hour after resistance training and cycling [42]. Our results were in line with the findings of Garwin et al. (1997) because no reduction was found in the state anxiety levels immediately after the exergame session [42]. As we did not evaluate the state anxiety levels one hour after the session, we cannot hypothesize about the possible outcomes.

Another aim of our study was to compare the exercise intensity between both the modes. We found no significant difference in HR and RPE between the single and multiplayer modes. We hypothesized that the multiplayer mode would elicit a higher exercise intensity. Exergames featuring a competitive element were more likely to induce a higher level of physical activity if played in a multiplayer mode rather than in a single-player mode [43]. McGuire and Willems (2015) compared intensity between the single and multiplayer modes of three modalities of the exergame Kinect Sports^®^ (soccer, boxing, track, and field) in young adults [44]. Ten young adults aged 23±5 years completed three modalities of the exergame Kinect Sports^®^ in Xbox 360 Kinect^®^. The intensity of the sessions was measured based on the HR. All the participants played the three modalities in single and multiplayer modes separated by a week’s interval. The results showed that the HR in the single-player mode was 120 ± 28 bpm (boxing), 82 ± 30 bpm (soccer), and 125 ± 21 bpm (track and field). At the same time, the HR in the multiplayer mode was 147 ± 18 bpm (boxing), 122 ± 15 bpm (soccer), and 136 ± 15 bpm (track and field). The authors found that the HR was higher during a multiplayer mode for boxing and soccer. This result indicated that playing exergame in the multiplayer mode can be more vigorous, but the effects are game-dependent [45]. Bonetti et al. (2010) compared acute exercise responses between conventional video gaming and isometric resistance exergame in the single and multiplayer modes on the HR and RPE [46]. They found no significant difference between single and multiplayer modes on the HR and RPE in conventional video gaming and an isometric resistance exergame. In the present study, no significant difference was found in the HR and RPE during the exergame Kinect Sports^®^ beach volleyball practice between the single and multiplayer modes. Our findings showed no significant difference in the mean HR and RPE between the single and multiplayer modes in healthy adult men.

In addition, previous exergame experience has been indicated as a factor that could influence exercise intensity since it minimizes the exertion and improves the movement precision [16]. Furthermore, the participants evaluated in the present study were physically active ($\dot{V}$O_2_max = ~52 mL/kg/min). This is a possible explanation that the exergame beach volleyball from Kinect Sports^®^ did not challenge the participant’s cardiorespiratory fitness level. A well-designed exergame will take people to “Flow Zones”, where the person’s abilities are matched as per a challenge, delivering feelings of pleasure and happiness. Thus, games should adapt to the players’ skills to keep them in the flow zone. However, if the challenge exceeds the person’s abilities, “anxiety” can be experienced, whereas minor challenges in terms of complexity and fun leads to “boredom” [46]. Therefore, the relationship between the challenge proposed by the exergame and the person’s skills are important. Thus, according to our results, the exergame Kinect Sports^®^ beach volleyball should be prescribed with caution to healthy adult men who seek to reach the recommended level of physical exercise proposed by the American College of Sports Medicine as light exercise intensity is evoked by this exergame [47].

Finally, our study compared the enjoyment levels between the two modes. Participants reported high levels of enjoyment in both single and multiplayer modes, without any significant differences. One possible explanation for the high enjoyment levels found in the present study may be related to the fact that young adults are more receptive to innovative technology [25]. Mackintosh et al. (2016) investigated the acute effect of the exergame Wii™ Boxing on enjoyment level in single vs. multiplayer modes in 36 university students. They found no significant differences in enjoyment between single and multiplayer modes [14]. Conversely, Peng and Crouse (2013) investigated the effects of a minigame Space Pop performed in the exergame Kinect Adventures^®^ on enjoyment levels between the single and multiplayer modes and found that the multiplayer mode presented with more enjoyment levels than the single-player mode [16]. These results indicate that the difference in single vs. multiplayer mode may be associated with the kind of game included for investigation or the duration of the intervention performed in studies (5 min or less). In the present study, no significant difference in the enjoyment levels associated with the exergame Kinect Sports^®^ beach volleyball was found between the single and multiplayer modes.

Our study has certain limitations. First, there was no control group characterized by quiet rest. Therefore, it is recommended that future studies include a non-exercise control group or placebo intervention. Second, due to our study design’s singularities, participants were not randomized to single or multiplayer modes. However, there were no statistical differences in participants’ characteristics between the two modes. Third, as we used questionnaires and scales, the results were dependent on the participants’ honesty. Fourth, we did not investigate any clinical population with diagnosed anxiety as a disorder, which might have affected the floor effect on decreasing state anxiety levels after both exergame modes. Sixth, due to the characteristics of the exergame used in our study, we could not control the magnitude of load and density (effort/pause ratio) in each game mode (singleplayer and multiplayer).

Finally, we investigated only adult male individuals. Nevertheless, we believe that these limitations do not limit our conclusions. Future research investigating different exercise intensities (light vs. moderate vs. vigorous) and modes (single vs. multiplayer) would be beneficial. They would provide more information about the exergame’s effect on state anxiety. It is also necessary to investigate the impact of the exergame Kinect Sports^®^ beach volleyball on the state anxiety levels in individuals with low cardiorespiratory fitness and/or anxiety disorders as well as participants from both sexes and different ages to better understand the impact of the exergames. 

## 5. Conclusions

There was no difference in the state anxiety levels between the single and multiplayer modes. Furthermore, both of the exergame modes did not evoke an anxiolytic effect. We did not find any significant difference between the enjoyment level and exercise intensity between the single and multiplayer modes.

## Figures and Tables

**Figure 1 ijerph-18-10957-f001:**
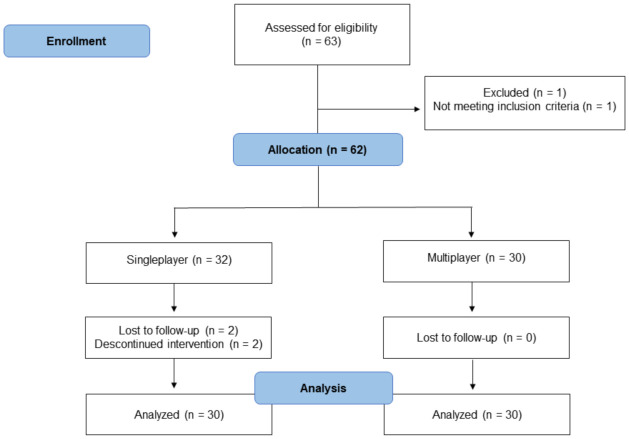
Flow diagram of the study.

**Figure 2 ijerph-18-10957-f002:**
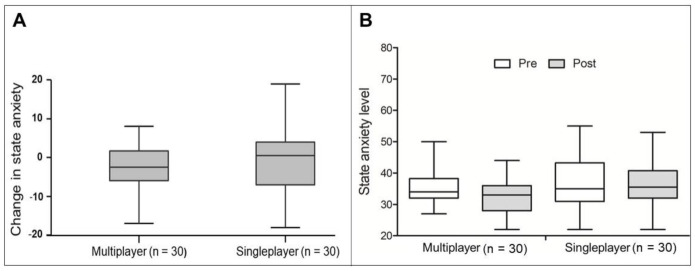
State anxiety and exergame. (**A**) Comparison of the state anxiety level changes after a session of the exergame Kinect Sports^®^ beach volleyball between the single and multiplayer modes. There is no significant difference between modes (*p*-value = 0.407). (**B**) Comparison of the state anxiety level before and after a session of the exergame Kinect Sports^®^ beach volleyball between the single and multiplayer modes. There is no significant difference in the single-player (*p*-value = 0.516) and multiplayer (*p*-value = 0.053) modes.

**Figure 3 ijerph-18-10957-f003:**
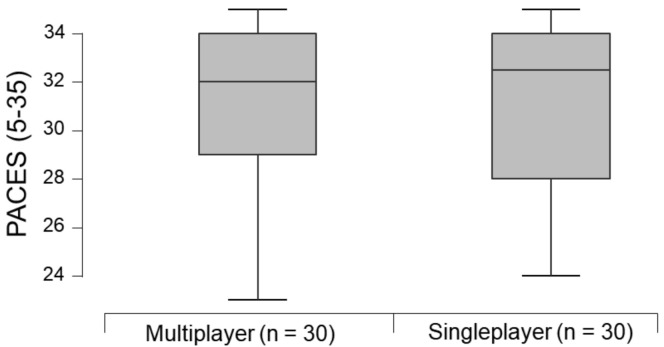
Comparison of the enjoyment level after a session of the exergame Kinect Sports^®^ beach volleyball between the single and multiplayer modes. There is no significant difference between modes (*p*-value = 1.000). PACES: Physical Activity Enjoyment Scale.

**Figure 4 ijerph-18-10957-f004:**
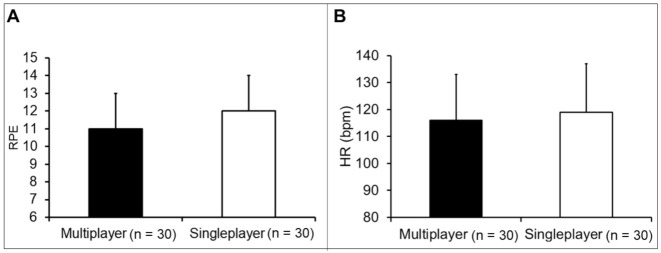
Exercise intensity and exergame. (**A**) Comparison of the RPE between the single and multiplayer modes after a session of the exergame Kinect Sports^®^ beach volleyball. There is no significant difference between modes (*p*-value = 0.201). Data are presented as mean ± standard deviation. RPE: the rating of perceived exertion. (**B**) Comparison of the HR between the single and multiplayer modes after a session of the exergame Kinect Sports^®^ beach volleyball. There is no significant difference between modes (*p*-value = 0.583). Data are presented as mean ± standard deviation. HR: Heart rate, Bpm: Beats per minute.

**Table 1 ijerph-18-10957-t001:** Characteristics of the participants.

	Singleplayer (n = 30)	Multiplayer (n = 30)	*p*-Value
	Median [IQR]	Median [IQR]	
Age (years)	22.30 [2.50]	21.90 [5.00]	0.900 ^a^
Body mass (kg)	72.45 [20.75]	77.70 [12.28]	0.412 ^a^
Height (m)	1.77 [0.12]	1.76 [0.09]	0.801 ^a^
Body mass index (kg/m^2^)	23.18 [6.42]	25.21 [3.86]	0.390 ^a^
Trait anxiety (20–80)	38.00 [13.00]	35.00 [13.00]	0.230 ^a^
HRmax predicted (bpm)	198.00 [3.00]	198.00 [2.00]	0.864 ^a^
Estimated $\dot{V}$O_2_max (mL/kg/min) ^c^	51.50 ± 7.00	51.80 ± 5.00	0.814 ^b^

IQR, interquartile range; HRmax, maximum heart rate; $\dot{V}$O_2_max: maximum oxygen uptake. ^a^ Mann–Whitney U test. ^b^ Independent Student’s *t* test. ^c^ Data presented as mean ± standard deviation.

## Data Availability

The data that support the findings of this study is available upon request to the corresponding author.

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
