# Peer review of "State Anxiety after Exergame Beach Volleyball Did Not Differ between the Single and Multiplayer Modes in Adult Men"

_ijerph, 2021, doi:10.3390/ijerph182010957_

Round 1

Reviewer 1 Report

The main limitatios is the study design and the intervention (one session and magnitude of load).

It is necessary to explain in greater detail the volume and density (effort/pause ratio) of the intervention (volleibal game used).

Author Response

Manuscript ID IJERPH-1387001

Title: State anxiety after exergame Beach Volleyball did not differ between the single and multiplayer modes in adult men

05-October-2021

Penny Gu

Managing Editor

Dear Editor,

                 We would like to thank the reviewers for their thorough review and insightful feedback; we have made all necessary revisions (highlighted in red in the manuscript) and answered the reviewers’ questions point-by-point. The manuscript has improved substantially, and we hope it is now suitable for publication in the International Journal of Environmental Research and Public Health.   

COMMENTS TO THE AUTHOR:

-------------------------------------------------------------------------------------------------------------------------------
Reviewer #1:

 Thank you for your review and constructive comments. We have provided a point-by-point answer for each one of your comment.

Method

The main limitatios is the study design and the intervention (one session and magnitude of load).

Answer:  Thank you for your insightful comment. In this study, we compared the acute effects between playing exergames in singleplayer vs. multiplayer mode on state anxiety. Due to the characteristics of the exergame applied in our study, it is not possible to monitor control the magnitude of load in each mode, because each session has particularities that depend on players. We have included this limitation in the revised manuscript in order to clarify and meet with your expectation.

Conversely, we evaluated exercise intensity by measuring heart rate and rating of perceived exertion and we found that exercise intensity can be classified as light according to American College of Sports Medicine’s guidelines.

It is necessary to explain in greater detail the volume and density (effort/pause ratio) of the intervention (volleibal game used).

Answer: Thank you for your constructive and insightful comment. The exergame Beach Volleyball has no time limit for each match. Thus, we limited the intervention time in approximately 30 minutes and the participants played around six matches with a short break between then (around 1-2 minutes). This period was necessary for participants to select the option to play another match on the videogame menu. This sentence was added in the revised version of manuscript in order to clarify. Regarding the density of the intervention, unfortunately we did not collect this data. Therefore, it was added as a study limitation in the revised manuscript.

Reviewer 2 Report

  1. line 14 (age: 21.98 [4.58] years..... 4,58 should be written ±4.58
  2. The statistical analysis is not very rigorous and tests of great interest are not carried out, with impractical results for coaches and professionals in the world of volleyball

  3. There aro too much figures. They may be unificated as they show some similar analysis.
  4. The references are not current at all. References from recent years must be added.

Author Response

Manuscript ID IJERPH-1387001

Title: State anxiety after exergame Beach Volleyball did not differ between the single and multiplayer modes in adult men

05-October-2021

Penny Gu

Managing Editor

Dear Editor,

                 We would like to thank the reviewers for their thorough review and insightful feedback; we have made all necessary revisions (highlighted in red in the manuscript) and answered the reviewers’ questions point-by-point. The manuscript has improved substantially, and we hope it is now suitable for publication in the International Journal of Environmental Research and Public Health.   

COMMENTS TO THE AUTHOR:

-------------------------------------------------------------------------------------------------------------------------------
Reviewer #2:

 Thank you for your review and constructive comments. We have provided a point-by-point answer for each one of your comment.

1.line 14 (age: 21.98 [4.58] years..... 4,58 should be written ± 4.58.

Answer: Thank you for your comment. Actually, these data presented a non-normal distribution, and therefore, we reported it as median and interquartile range. We have reworded this sentence in the revised manuscript in order to clarify.  

2.The statistical analysis is not very rigorous and tests of great interest are not carried out, with impractical results for coaches and professionals in the world of volleyball.  

Answer: Thank you for your comment. The test initially selected was the Two-Way ANOVA with factors between groups. However, the requirements for this parametric test were not met, so segmentation was necessary (Goss-Sampson, M. A. (2020). Statistical Analysis in JASP 0.14: A Guide for Students. November 2020). Please let us know if this explanation does not resolve your doubts in this matter.

Moreover, we are positive that our study can contribute to the general population and/or health providers who searches for alternative way to maintain regular physical activity and, consequently, to improve health status. Please let us know if this explanation does not resolve your doubts in this matter.

3.There aro too much figures. They may be unificated as they show some similar analysis.

Answer: Thank you for calling our attention. We have unified the Figures 2-3, and the figures 5-6 as suggested by you. In total, as resulted of this change, there are four figures. Please let us know if these changes do not resolve your doubts in this matter.

4.The references are not current at all. References from recent years must be added.

Answer: Thank you for your comment. We replaced references “5”, “30” and “35” per recent ones. However, it is worth mentioning that some classical references represent the original articles about the methods and statistical analysis applied in our study.

Round 2

Reviewer 1 Report

No comments

Reviewer 2 Report

Thank you for your explanations. Everything is much clear now.